# Acquisition of Immune Privilege in GBM Tumors: Role of Prostaglandins and Bile Salts

**DOI:** 10.3390/ijms24043198

**Published:** 2023-02-06

**Authors:** Martyn A. Sharpe, David S. Baskin, Ryan D. Johnson, Alexandra M. Baskin

**Affiliations:** 1Kenneth R. Peak Brain and Pituitary Tumor Treatment Center, Department of Neurosurgery, Houston Methodist Neurological Institute, Houston Methodist Hospital and Research Institute, Houston, TX 77030, USA; 2Department of Neurological Surgery, Weill Cornell Medical College, New York, NY 10065, USA; 3Department of Natural Science, Marine Science, Hawaii Pacific University, Honolulu, HI 96801, USA

**Keywords:** glioblastoma (GBM), regulatory T-cells (Tregs), microglia, prostaglandins, bile, sperm, myeloid-derived suppressor cells, tumor-associated macrophages

## Abstract

Based on the postulate that glioblastoma (GBM) tumors generate anti-inflammatory prostaglandins and bile salts to gain immune privilege, we analyzed 712 tumors in-silico from three GBM transcriptome databases for prostaglandin and bile synthesis/signaling enzyme-transcript markers. A pan-database correlation analysis was performed to identify cell-specific signal generation and downstream effects. The tumors were stratified by their ability to generate prostaglandins, their competency in bile salt synthesis, and the presence of bile acid receptors nuclear receptor subfamily 1, group H, member 4 (NR1H4) and G protein-coupled bile acid receptor 1 (GPBAR1). The survival analysis indicates that tumors capable of prostaglandin and/or bile salt synthesis are linked to poor outcomes. Tumor prostaglandin D_2_ and F_2_ syntheses are derived from infiltrating microglia, whereas prostaglandin E_2_ synthesis is derived from neutrophils. GBMs drive the microglial synthesis of PGD_2_/F_2_ by releasing/activating complement system component C3a. GBM expression of sperm-associated heat-shock proteins appears to stimulate neutrophilic PGE_2_ synthesis. The tumors that generate bile and express high levels of bile receptor NR1H4 have a fetal liver phenotype and a RORC-Treg infiltration signature. The bile-generating tumors that express high levels of GPBAR1 are infiltrated with immunosuppressive microglia/macrophage/myeloid-derived suppressor cells. These findings provide insight into how GBMs generate immune privilege and may explain the failure of checkpoint inhibitor therapy and provide novel targets for treatment.

## 1. Introduction

Our prior analysis of glioblastoma (GBM) transcriptomes highlighted the expression of genes involved in the syntheses of estrogens and androgens. These steroids create immunosuppressive niches and allow GBMs to express reproduction-related proteins that evoke immune privilege [1].

The gastrointestinal (GI) tract is another system that grants immune privilege to ‘non-self’ antigens and epitopes. Our bodies digest many foreign proteins in the presence of microflora, and without some form of immune evasion, there could be off-target inflammation and immune responses during digestion [2]. Prostaglandins and bile salts play essential roles in producing immune evasion in the GI tract, thus preventing inappropriate immune attacks [3,4,5,6,7].

The objective of this study is to investigate the role of prostaglandins and bile acids in sculpting immunosuppressive environments in GBM tumors. The synthesis of prostaglandins by different tumor types and their effects within the tumor microenvironment have been covered in depth by Jara-Gutiérrez and Baladrón [8]. Prostaglandins are a group of hormone-like eicosanoids that modulate cell function, especially in immune cells.

We focus on a subset of these compounds: prostaglandins E_2_ (PGE_2_), D_2_ (PGD_2_), and F_2_ (PGF_2_). Figure 1A shows the synthetic routes for these prostaglandins and their inactivation by the catabolic enzyme hydroxyprostaglandin dehydrogenase (HPGD). All prostaglandins originate from the action of the cyclooxygenases COX1 and COX2 (PTGS1 and PTGS2) on arachidonic acid to generate the intermediate prostaglandin H_2_ (PGH_2_).

**PGD_2_:** PGD_2_ is the most abundant prostaglandin released from neuronal and non-neuronal cells, including microglia and macrophages. PGD_2_ is synthesized from PGH_2_ by two PGD_2_ synthases: lipocalin-type PGD_2_ synthase (L-PGDS or PTGDS) and hematopoietic PGD_2_ synthase (HPGDS); the latter is a microglial marker [9,10]. Microglial HPGDS-derived PGD_2_ is elevated by ∼23-fold following the mobilization of intracellular Ca^2+^ with ionophore A23187 [11]. Microglial HPGDS-derived PGD_2_ is vasodilatory, increasing brain blood flow and acting on endothelial cell PGD_2_ receptor [12]. In addition, PGD_2_ aids inflammation resolution and drives macrophage polarization into the anti-inflammatory M2 state [13]. Part of the physiological brain signaling is the release of complement component C3 by astrocytes, which, when mature, binds to microglial C3AR1 receptors, causing the synthesis of PGD_2_ [11]. The complement component CFB is the main route for converting inactive C3 into active C3a in the brain [14]. Microglial C3AR1 receptor is also activated by the neuropeptide TLQP-21, a C3a agonist [15]. C3AR1 activation causes cAMP signaling inhibition, elevates cytosolic Ca^2+^, and activates phospholipase C, STAT3, and the extracellular signal-regulated kinase 1/2(ERK1/2) pathway [16]. In medulloblastoma tumors, the crosstalk between C3a and C3RA1, which is expressed by infiltrating astrocytes and microglia, causes the release of Tumor necrosis factor alpha (TNF-α) [17], and TNF-α stimulates chemokine ligand 5 (CCL5) release in glia and macrophages [18]. Antagonism of microglial chemokine receptor 5 (CCR5) inhibits M2 > M1 polarization [19], and there is much evidence of glioma CCL5/microglial CCR5 crosstalk in GBM tumors [20,21]. Physiologically, the CCL5/CCR5 axis is a feature of wound healing [22,23], and this appears to be subverted by tumors [24,25,26,27].

**PGF_2_:** Aldo-Keto Reductase Family 1 Member C3 (AKR1C3) generates angiogenic prostaglandin PGF_2_ from PGH_2_ and PGD_2_ [28,29]. AKR1C3 and the PGF_2_ receptor, PTGFR, are highly expressed during the proliferative phase of the menstrual cycle, which is characterized by endometrial cell proliferation and angiogenesis [30]. Both are also expressed in high levels in the placenta [30,31,32,33,34]. Human microglia also express AKR1C3 and synthesize PGF_2_ [35], and activation of pericyte PTGFR causes the expression of trophoblast glycoprotein (TPBG). TPBG coordinates pericyte migration and angiogenic activities [36,37,38,39,40,41]. PGF_2_ signaling by PTGFR attracts circulating angiogenic stem cells via the C-X-C motif chemokine ligand 12 (CXCL12) and C-X-C motif chemokine receptor 12 (CXCL12) axis and is widely utilized in fetal organ development [38,40,41,42]. In addition, microglia and astrocytes express CD38 [43], which potentiates the angiogenic action of CXCL12 [36,44]. PGF_2_ receptor negative regulator (PTGFRN) is an inhibitor of PGF_2_ receptor signaling and inhibits PTGFR. PTGFRN is found in various cancers, including aggressive GBM [45,46]. PTGFRN is elevated in GBM tumors with a hypoxic phenotype, with its expression modulated by hypoxia-responsive microRNA, miR-137 [47,48].

**Control of PGD_2_/PGF_2_ ratios:** Since hypoxia/hypoxia-inducible factor (HIF) signaling downregulates AKR1C3 expression, PGD_2_/F_2_ ratios should correlate with hypoxic/normoxic conditions [49,50]. Estrogen/estrogen receptor-β signaling elevates AKR1C3 expression, and it has been previously shown that estrogenic tumors are heavily infiltrated by M2-polarized microglia and macrophages [1,31]. Estrogen is known to elevate CCL18 secretion in M2 macrophages [51], but it downregulates the expression of complement components, including C3, in normal brain tissue [52]. A recent study demonstrated that CCL18 released by GBM-infiltrating microglia/macrophages promotes glioma growth [53].

**PGE_2_:** PGE_2_ is synthesized from PGH_2_ by three different prostaglandin E_2_ synthase enzymes that are the products of PTGES, PTGES2, and PTGES3 genes [54]. Tumor-associated neutrophils expressing PTGS2 and PTGES can contribute to poor outcomes in GBM xenograft models [55,56]. In addition, seminal PGE_2_ drives the process by which the female immune system awards immune privilege to seminal antigens/epitopes [1,57]. Human sperm is coated with immunosuppressive heat-shock proteins (HSPs), epididymal secretory sperm-binding protein (HSPA6), HSP90B1, and DNAJ Heat Shock Protein Family (Hsp40) Member B1 (DNAJB1) [58,59]. These activate epithelial cells Toll-like Receptor 2 and 4 (TLR2&4), thereby stimulating PGE_2_ and CXCL8 production [60]. In colorectal cancer cells, the expression of HSPA6 emulates the reproductive immunomodulatory action of sperm signaling [61]. Neutrophils generate PGE_2_ when exposed to sperm [62], thereby generating an immunosuppressive environment [63] that results from the co-stimulation of TLR2/4 receptors [60,64]. Evidence for the downstream effects of PGE_2_ signaling may be observed from the increases in the transcript levels of neutrophilic ARNTL [65,66], stromal/T-cell GADD45A [67,68], and astrocytic CEBPD [69].

**Bile Salts and Receptors:** Primary bile acids, cholic acid (CA), and chenodeoxycholic acid (CDCA) are generated by the adult liver, predominantly using the ‘classical’ pathway, which is rate limited by the activity of the enzyme cholesterol 7α-hydroxylase (CYP7A1) [70]. However, in the brain and fetal liver, synthesis is driven by the ‘acidic’ bile synthesis pathway, where sterol 27-hydroxylase (CYP27A1 gene product) is rate limiting [71].

CYP27A1 is a GBM oncogene whose expression is downregulated by the tumor suppressor miR-204 [72,73]. Sterol 27-hydroxylase acts on cholesterol to generate 27-hydroxycholesterol (27-HC). 27-HC is immunosuppressive in vitro and in vivo [74], and elevated serum 27-HC correlates with poor GBM patient outcome [75].

27-HC is a substrate for the enzymatic products of the genes CYP7B1 and CYP8B1, leading to the synthesis of the primary bile salts CDCA and CA, respectively. ACOX2, a branched-chain acyl-CoA oxidase, is also required for the final steps of primary bile salt production and has been identified as a high-risk gene in the progression of GBM [76].

In the context of immune signaling, these primary bile salts are anti-inflammatory. They can activate a pair of receptors: the plasma membrane-spanning G protein-coupled bile acid receptor 1 (GPBAR1) and the nuclear receptor subfamily 1, group H, member 4 (NR1H4), Figure 1B. These bile receptors are expressed in immune system cells, including microglia, macrophages, dendritic cells, and T cells. The activation of either of these receptors produces an anti-inflammatory phenotype [7]. Although bile salt immune modulation is mainly associated with the gut, bile signaling is also part of reproductive biology. In males, bile salt activation of testicular GPBAR1 [77] and NR1H4 [78] in specific cell sets is vital for normal testicular stem cell hemostasis. A female’s bile-rich follicular fluid released during ovulation drives immunosuppression, readying the uterus for implantation [79].

Activation of macrophage GPBAR1 causes Interleukin 1 Beta (IL1B) release [80], and macrophage-derived IL1B causes breast cancer cells to release CCL13 [81]. In GBM tumors, the release of IL1B by tumor-associated macrophages (TAM’s) causes gliomas to release CCL13, driving the recruitment of monocytes and their transformation into TAMs, which then drive progression [82]. It is noteworthy that the production of CCL13 is potentiated by estrogen [83]. 

## 2. Results

### 2.1. Prostaglandin Sculpting of the Tumor Immunological Phenotypes

Figure 2A shows the median levels of selected transcripts in the four prostaglandin tumor populations to validate the stratification methodology. 

The chi-squared statistic below each pie chart demonstrates the stratification procedure employed, which captures the desired transcript levels. Figure 2B shows the canonical divisions of GBM sub-types with the four prostaglandin phenotypes, which have shown statistical significance, and the chi-squared statistics. The classical phenotype is associated with a tumor’s ability to synthesize PGE_2_, whereas prostaglandin impoverishment is associated with the proneural phenotype.

As the expression levels of AKR1C3 were used to stratify the tumors into PGD_2_ and PGF_2_ groups, correlation analysis was performed to validate this procedure in three different transcriptome databases, as shown in Appendix A (left). AKR1C3 expression in the GBM tumors is inversely correlated with the hypoxia marker ADM and PGF_2_ receptor negative regulator (PTGFRN), an inhibitor of PGF_2_ receptor signaling. Furthermore, the expression of the PGF_2_ receptor, PTGFR, is highly correlated with TPBG and CXCL12, as shown in Appendix A (right). These data suggest a normoxic PGF_2_-generating phenotype with PTGFR-driven angiogenic TPBG/CXCL12 axis.

Correlation analysis can be used to identify which cell types could be responsible for prostaglandin synthesis. In Appendix A, the plots obtained directly from Gliovis, which show the correlation with representative markers of microglia adenosine A3 receptor (ADORA3) and neutrophils peptidase inhibitor 3 (PI3), are presented. The PGH_2_-synthesizing enzyme gene PTGS1 (COX1) and the PGD_2_-synthesizing gene HPGDS are cross correlated with the microglial marker ADORA3. The PGH_2_-synthesizing enzyme gene PTGS2 (COX2) and the PGE_2_-synthesizing gene PTGES are cross correlated with the neutrophil marker PI3. Thus, tumor PGD_2_ synthesis is microglial via COX1, whereas PGE_2_ synthesis is neutrophilic, driven by COX2.

The Kaplan–Meier survival curves, median survival, and statistical significance of the cohorts stratified into the four prostaglandin phenotypes are presented in Figure 3A, using data from the three GBM patient databases.

In all three GBM patient populations, the overall survival of the ‘Low PG’ group is highest. Although the ability of tumors to synthesize either PGF_2_ or PGE_2_ is detrimental to patient outcomes, tumors with the ability to generate PGD_2_ are the most lethal.

Figure 3B highlights the tumor/immune cell phenotype of the combined PGD_2_/PGF_2_ stratified groups compared to the PGE_2_ and the ‘Low PG’ groups. The pie charts represent the median levels of transcripts of the three groups. There are 25% more microglia in the PGD_2_/PGF_2_ group than in the ‘Low PG’ group. Conversely, in the PGE_2_ group, there are 10% fewer microglia than in the ‘Low PG’ group. Microglia have transcriptome signatures that indicate both classical activation and NF-κβ signaling pathways. In addition to microglia, the tumors of the PGD_2_/PGF_2_ group have elevated levels of both myeloid-derived suppressor cells (MDSCs) and TAM gene expression markers. In contrast, TAMs, not MDSCs, are elevated in the PGE_2_ group and are highly M2 polarized. The final four pie charts in Figure 3B indicate that the PGD_2_/PGF_2_ phenotype is associated with the CCL5/CCR5 axis. Evidence for the PGD_2_/PGF_2_ phenotype being driven by microglial C3AR1 activation via tumor-expressed CFB/C3 generating C3a is found by examining the phenotypical differences between the tumors that generate PGD_2_, PGF_2_, and PGE_2_, as shown in Figure 4.

Evidence that GBM emulates astrocyte behavior by generating C3a and causing microglia to synthesize PGD_2_/PGF_2_ was sought. Using a pair of transcriptome databases, the correlation analysis shows the microglial associations with significant levels of specific complement receptors (Appendix A). Microglial marker ADORA3 correlates highly with microglia/macrophage receptors C3AR1 and C5AR1. In addition, microglial marker ADORA3 correlates highly with complement components C3, CFB, and CFD. Appendix A shows that microglial PGD_2_-synthesizing enzyme transcripts PTGS1 and HPGDS correlate highly with microglial/macrophage transcription factor IRF8. 

The tumors stratified as expressing PGD_2_ or PGF_2_ have an elevated expression of C3AR1, C3, and CFB. The correlation analysis, drawing from four databases, demonstrates that microglial-specific marker ADORA3 is more closely associated with PTGS1 and C3AR1, whereas macrophage marker CD68 is better associated with PTGS2 and C5AR1 (Appendix A).

Utilizing hypoxia-regulated gene adrenomedullin (ADM) transcripts to represent HIF-signaling, the PGD_2_ and PGE_2_ groups are hypoxic, while the PGF_2_ sub-population is normoxic.

The PGF_2_ tumor phenotype has elevated CCL2, IL6, IL1A, and IL1B, all of which are secreted by uterine smooth muscle cells treated with PGF_2_. In addition, CD38 and CXCL12 are angiogenic [44,85,86], acting on pericytes and perivascular fibroblasts [87,88,89], and both are found in the PGF_2_ phenotype. The correlation analysis indicates that PTGFR is correlated with makers of perivascular fibroblasts, angiogenic mesenchymal stem cells fibulin-1 (FBLN1), fibrillin-1 (FBN1), glypican-3 (GPC3), receptor tyrosine kinase like orphan receptor 2 (ROR2), and smooth muscle marker slit guidance ligand 3 (SLIT3) (Appendix A).

### 2.2. Evidence for Sexual Hijacking in the PGE_2_ Phenotype

The PGE_2_ group displays upregulation of classical GBM markers: PDGFA, VEGFA, EGFR, and SOX9 [90,91,92,93]. Three sperm HSPs are highly expressed in the PGE_2_-producing tumor group (Figure 4). The correlation analysis shows a correlation between these HSPs with COX2 and neutrophil markers PI3 and chitinase-3-like protein 1 (CHI3L1) (Appendix A). HSPA6 expression, PTGS2, PTGES1, CCAAT/enhancer-binding protein delta (CEBPD) , neutrophil marker CHI3L1, and transforming growth factor beta 1 (TGFβ1) are all highly correlated (Appendix A). 

The bottom row of Figure 4 demonstrates that three markers of PGE_2_ signaling are upregulated in the HSP/neutrophil/PGE_2_ tumor group—neutrophilic aryl hydrocarbon receptor nuclear translocator-like protein 1 (ARNTL), stromal/T-cell growth arrest and DNA-damage-inducible protein GADD45 alpha (GADD45A), and astrocytic CEBPD.

### 2.3. Bile Salt Sculpting of the Tumor Immunological Phenotypes

The transcriptome analysis shows that the levels of CYP7A1, the master controller of the classical bile synthesis pathway, are lower in GBM than in surrounding brain tissue. 

However, this is not the case for the enzymes of the bile ‘acidic’ pathway, including CYP27A1, ACOX2, and 3 beta-hydroxysteroid dehydrogenase type 7 (HSD3B7) [94].

In Appendix A, utilizing four different GBM transcriptome databases, it is shown that CYP27A1 and ACOX2 enzymes are overexpressed in GBM, and that overexpression is linked to poor patient outcomes (data taken directly from Gliovis).

When the GBM tumors were stratified based on the ability to synthesize bile by utilizing the expression levels CYP27A1 and ACOX2, a ‘Low Bile’ population was identified in only 18% of the tumors. The bile-generating tumors could not be divided into two populations based on the expression of the bile receptors GPBAR1 and NR1H4, as some tumors expressed both receptors at high levels. These bile-synthesis positive tumors were, therefore, divided into three equally sized groups: high GPBAR1 (NR1H4)_LOW_, high NR1H4 (GPBAR1)_LOW_, and ‘Mixed’ (Materials and Methods). This stratification process captures the overexpression of CYP27A1 and ACOX2, with the chi-squared tests showing *p* < 10^−7^, as shown in Figure 5A. Figure 5B represents the canonical divisions of the GBM sub-types within the four-bile acid sub-populations, with the chi-squared statistics included where relevant. 

The classical phenotype is associated with the ‘mixed’ bile receptor sub-type, the mesenchymal phenotype is associated with tumors rich in GPBAR1, and the low bile synthesis subgroup is associated with the proneural phenotype.

The Kaplan–Meier survival analysis indicates that the patients hosting tumors of three bile receptor subgroups have far worse outcomes than the patients with tumors lacking the expression of bile synthesis genes, as shown in Figure 6A. Figure 6B illustrates that the stratification procedure captures a Treg-driven and androgenic phenotype in the NR1H4 tumor population (Red). Firstly, the NR1H4-rich tumors have highly elevated expression levels of adult and fetal liver markers. Secondly, in these NR1H4-rich tumors, markers of RAR Related Orphan Receptor C (RORC)-rich Tregs (RORC-Tregs) are elevated along with the levels of RORC-Treg target mucin, MUC16. A marker of NR1H4 signaling, bile-sensitive hormone fibroblast growth factor 19 (FGF19), is also highly expressed in this group. The pie charts show that the NR1H4-rich tumors are androgenic, based on the median levels of an androgen marker, keratin 37 (KRT37).

In contrast, the GPBAR1 receptor-rich tumors (Blue) are estrogenic, clustering with estrogen reporter genes, such as vav guanine nucleotide exchange factor 1 (VAV1). Tumors enriched within the GPBAR1 are more heavily infiltrated with microglia, TAMs, and MDSCs, than tumors enriched with NR1H4, Figure 7. This corresponds to the previously observed enrichment in estrogenic tumors [1].

In the GPBAR1-expressing subtype, there are high levels of M2 polarization markers. The GPBAR1-expressing subtype shows a distinct overlap with the microglial C3/C3AR1-driven PGD_2_/PGF_2_ phenotypes, including evidence for an active CCL5/CCR5 axis. Members of the IL1 family, including IL1B, IL33, and IL18, are upregulated in the GPBAR1 tumors and are generally identified as contributors to tumor immunosuppression.

Interferon regulatory factors 4 and 8 (IRF4 and 8) have a reciprocal relationship with the two bile acid receptors, with IRF8 high in the GPBAR1-rich tumors and low in those with high levels of NR1H4. In contrast, IRF4 expresses the opposite pattern, Figure 7. Utilizing a pair of GBM transcriptome databases, the results of the cross-correlation analysis indicating that IRF4 correlates with markers of RORC-Tregs and that IRF8 expression correlates with microglia can be found in Appendix A. 

Contingency tables were prepared, and cross-correlation analysis was performed to examine the overlap between the tumors stratified with respect to bile-receptor signaling compared with the tumors stratified by hypoxia/Treg infiltration and steroidogenesis, as previously described [1], and prostaglandin synthesis (Appendix A). The tumors identified as belonging to the NR1H4 sub-population are over-represented with RORC-Treg infiltration. Furthermore, these NR1H4-rich tumors are also over-represented with the previously identified androgenic subgroups but do not overlap with any prostaglandin-stratified group.

The tumors identified as belonging to the GPBAR1 sub-population are over-represented in hypoxic tumors with low RORC-Treg infiltration and with the estrogenic (low androgen) phenotype. The patient populations with low prostaglandin or low bile/bile receptor phenotypes, which have the best patient outcomes, show statistically significant overlap (Appendix A).

## 3. Discussion

We have demonstrated that one GBM tumor phenotype displays microglial markers and components of the complement system, which are known to drive prostaglandin D_2_ and F_2_ microglia. Another GBM tumor phenotype rich in sperm-associated, toll-like receptor 4 ligands is correlated with neutrophil PGE_2_ synthesis. Furthermore, some GBMs emulate the fetal liver’s genetic signature, allowing them to generate bile salts via the fetal ‘acidic’ synthesis pathway. Our data suggest that microglia are the primary source of PGD_2_/PGF_2_, and neutrophils are a significant source of PGE_2_.

### 3.1. Androgens, Estrogens, Prostaglandins, and Bile

Cancer cells can increase their survival by subverting the immune system in many ways, including the well-characterized ability of IDH-mutants to generate immunosuppressive oncometabolite (D)-2-hydroxyglutarate and elevate immunosuppressive ROS [95,96]. The roles of estrogen and androgen generation by GBM tumors in utilizing immunosuppressive effects of reproductive biology were previously explored [1]. The same steroid core synthesizes bile salts, which are profoundly immunosuppressive in the brain, the gut, and reproductive biology. The ability of tumors to generate bile salts to target immune cell receptors is another means of immune evasion. Prostaglandin synthesis in GBM tumors appears to be due to another immune subversion strategy, with tumors emulating astrocytes to generate complement component C3a, coaxing microglia to synthesize PGD_2_/PGF_2_, or emulating sperm coating of HSPs to have neutrophils generate immunosuppressive PGE_2_. Tumors that express sperm coating of HSPs also express high levels of SOX9. GBM expression of SOX9 is associated with ‘stemness’ [97,98,99]; it functions as a ‘masculinizer’ in germ cells [100].

Previous knockout studies have demonstrated that astrocytes and microglia are unaffected by bile/NR1H4 signaling [101]. Bile salts modulate RORC-Tregs [6], a population previously identified in GBM tumors as being associated with androgen signaling [1]. Primary bile acids bind to NR1H4 in RORC-expressing T cells, driving Teff > Treg anti-inflammatory polarization [5,6]. This same tumor group expresses AFP, and AFP binds estrogen, but not androgen, and blocks the estrogen-driven signaling [102]. Additionally, AFP has been shown to drive fetal and adult liver cancer development, potentiating FGFR and transmembrane mucin signaling [103]. MUC16 levels are highly upregulated in NR1H4 tumors, and its proteolysis-derived peptidyl fragment, CA-125, generates/activates reproductive Tregs in males and females [104,105].

### 3.2. Adult vs. Fetal Bile Salt Synthesis

Most bile synthesis occurs through the ‘classical’ pathway in the adult liver. In adults, the expression of the rate-controlling enzyme of the classical pathway, CYP7A1, determines the level of bile synthesis [2,106,107]. Classical bile synthesis is under feedback control, with enterocytes in the lower gut monitoring bile levels via NR1H4. When bile levels are high, these cells release the hormone FGF19. Circulating FGF19 binds to the receptor FGFR4 of liver cells, signaling a downregulation of CYP7A1 expression and, thus, dropping the rate of bile acid synthesis. Adults cannot support high levels of bile synthesis in FGF19 [71,108]. In contrast, the livers of neonates generate primary bile acids using the acidic bile synthesis pathway, where the CYP27A1 gene product is the most rate-controlling enzyme [71,107,109]. In neonates, CYP7A1 activity is undetectable prior to 30 weeks of gestation. This co-incidence of bile synthesis by the acidic pathway and elevation of FGF19 in a subset of GBM tumors is observed in neonates and preterm infants [71,108].

### 3.3. Diagnosis, Tracking, and Treatment of GBM with Vaccines

A large fraction of GBMs express testicular-specific proteins [1] and/or fetal liver-specific transcripts. Serum fetal liver proteins are routinely measured in hospitals to diagnose pregnancy-linked pathologies, i.e., α-fetoprotein or placental alkaline phosphatase and pregnancy-specific beta-2, 6, and 8. These validated tests could also be used in GBM patients for tracking tumor response to treatment and progression. Human fetal liver synthesizes and secretes biologically active chorionic gonadotropin. Provisional ELISA assays show that this pregnancy marker protein is present in the serum of both male and female GBM patients.

It has previously been demonstrated that GBM patients have circulating antibodies toward testicular-specific proteins [1]. The expression of fetal-specific proteins by GBM tumors opens an extensive range of new vaccination targets. Their efficacy will be improved if it is demonstrated that GBM patient serum has pre-existing antibodies toward fetal proteins. The advantage of targeting a fetal protein with an anti-cancer vaccine is the lower likelihood of off-target autoimmune responses.

## 4. Materials and Methods

### 4.1. GBM Transcriptomes

The in-silico transcriptome analysis was derived from three publicly available datasets using only IDH wild-type GBM tumors obtained during the first resection: the TCGA/U133 GBM microarray dataset (389 patients), the Agilent microarray (201 patients), and the Gravendeel [110] database (122 patients). In addition, the transcriptome GBM sub-type classifications and the patient survival data were imported from the cBioPortal [111] and Gliovis platforms [112]. Appendix A contain the plots and statistics from the Gliovis platform.

### 4.2. Stratification of GBM Phenotypes

#### 4.2.1. Normalization of Transcript Levels

When a value of N/A was returned for a transcript level in a dataset, it was assigned a median transcript level. Then, each dataset’s absolute transcript levels were normalized to 0–1. Next, tumors were stratified independently on each dataset based on the transcript levels. Finally, the stratified data were combined, returning *n* = 712 (wild-type IDH) tumors obtained at first resection.

#### 4.2.2. Stratification of GBM Tumors by Prostaglandins

The patient GBM tumor transcriptome data were stratified into four groups based on the ability or inability to synthesize the prostaglandins PGD_2_, PGF_2_, and PGE_2_, as shown in Figure 8A. The ‘Low PG’ tumor phenotype was obtained by ranking the transcript levels to identify those tumors that were PTGS1_LOW_ and PTGS2_LOW_, and/or HPGD^HIGH^. PGE_2_-generating tumors were AKR1C3_LOW_, PTGDS_LOW_, and HPGDS_LOW_ and comprised the prostaglandin-synthesizing tumors. The remaining PGD_2_/PGF_2_ tumors were divided into PGD_2_ tumors (AKR1C3_LOW_) and a PGF_2_ synthesizing group (AKR1C3^HIGH^).

#### 4.2.3. Stratification of GBM Tumors by Bile Synthesis and Receptors

Figure 8B outlines the methodology used to stratify the tumors based on the ability to synthesize bile and based on the receptors for bile. The tumors lacking CYP27A1 or ACOX2, or lacking both AKR1C3 and AKR1D1, were defined as part of the ‘Low Bile’ subgroup. In a two-step process, the remaining tumors were divided into three equally sized populations based on their rank when sorted by the NR1H4/GPBAR1 ratio. The upper 50% of the tumors sorted by the NR1H4/GPBAR1 ratio were stratified by the NR1H4 levels, generating an NR1H4^HIGH^ (NR1H4/GPBAR1)^HIGH^ population, or the NR1H4 group. Likewise, the lower 50% of the tumors ranked by the NR1H4/GPBAR1 ratio were stratified to produce a GPBAR1^HIGH^ (NR1H4/GPBAR1)_LOW_ population, or the GPBAR1 group. The remaining tumors were defined as ‘Mixed’ due to the expression of a mixture of both bile acid receptors.

### 4.3. Estimation of Infiltration by Immune Cells, Polarization, and Reproduction-Related Markers

As reported in our previous manuscript [1], the ‘gene basket’ approach was used.

#### 4.3.1. Microglia Levels and Activation Status

Total microglial levels were calculated from the average of normalized ADORA3, IGSF6, TBXAS1, SASH3, and P2RY13 transcripts. Inflammatory status was calculated by dividing total microglia by the averaged levels of genes known to be downregulated during an inflammatory response toward lipopolysaccharide: P2RY12, TMEM119, and GPR34 [1,113]. NF-κB activation status was determined by calculating the average of five genes elevated by its activation (GCLC, NQO1, GCLM, NFKBIA, and SLC39A8) and by the average of three genes downregulated by NF-κB (ZDHHC22, BCL7A, and GNG4) [9,10,113]. In the cross-correlation analysis, the linkage of microglial/macrophage transcription factor IRF8, a marker of microglial activation, was utilized [114,115].

#### 4.3.2. Infiltration of Tumors by Macrophages, MDSCs, Neutrophils, and Tregs

Macrophage infiltration was calculated from the average of the markers ITGAM and CD68 and the M2/M1 polarization from the ratio of CD163 by ITGAX. Myeloid cell marker CD33 was used to estimate the levels of MDSCs. Neutrophil levels were calculated from the averaged PI3 and CHI3L1 levels. Tumor RORC-Treg infiltration was calculated from the averaged, normalized levels of FOXP3, CTLA4, GITR(TNFRSF18), RORC, and GATA3 [1]. Resident mucosal RORC-Tregs express IRF4, PRDM1, and IL10 [116]. IRF4 expression drives Teff > Treg in tumor-infiltrating T cells [117,118] via PRDM1 to generate IL10-expressing Tregs [119]. IRF8 is a significant controller of microglial and macrophage activation [114,115,120]. IRF8 causes an upregulation of SPI1 [89] and a downregulation of TCF4 [121] and is critical for microglial expansion and activation [120,122]. These two expression factors may be used in a cross-correlation analysis to differentiate between Treg and microglial-rich tumors.

#### 4.3.3. Identification of Vascular Markers in Tumors

In the cross-correlation studies, the data from the human brain vascular atlas were used to identify markers of perivascular fibroblasts (FBLN1, FBN1, GPC3, and ROR2) [123], and angiogenic mesenchymal stem cell/smooth muscle was identified by utilizing SLIT3 [124,125].

#### 4.3.4. Reproduction-Associated Markers and Proxies

The averages of normalized transcripts of THBD, THEMIS2, SERPINA1, PIK3CG, and VAV1 were used as an estrogen response proxy. Androgen levels were estimated using the transcript levels of KRT37 as a proxy [126]. Sperm-associated HSP levels were calculated by averaging the three normalized transcripts of sperm HSPs, including HSPA6, HSP90B1, and DNAJB1 [58,59]. The averages of the hypoxia-response gene transcripts, SLC2A1, SLC2A3, VEGFA, VLDLR, and ADM, were used as a hypoxia proxy [1].

### 4.4. Statistical Analyses

Two different statistical tests were used to determine if the stratification procedures generated four different tumor phenotypes, as shown in Figure 2A and Figure 5A.

Firstly, the median enzyme/receptor transcript levels in the four ‘PG’ or ‘Bile’ groups were tested against the expected global population using chi-squared tests to answer the research question: Are the transcript medians of the four groups different from the population from which they were taken?

Secondly, the distribution of the enzyme/receptor transcript levels within each of the prostaglandin or bile synthesizing groups was compared with their distribution within the respective ‘Low PG’ or ‘Low Bile’ group using *t*-tests to answer the research question: Is the distribution of the transcript levels in the PG-positive or Bile-positive groups different from the respective ‘Low PG’ or ‘Low Bile’ groups?

Each patient tumor was identified as belonging to one of the three canonical GMP subtypes (classical, mesenchymal, or proneural) independently using the algorithms described on the Gliovis platform. Then, utilizing the contingency tables, chi-squared tests were performed to analyze the relationships between the distribution of these three canonical classifications within each of the four stratified ‘PG’ or ‘Bile’ groups and the global population distribution (Figure 2B and Figure 5B).

The distribution of cell-type proxies and individual gene transcripts were tested for statistical significance for each of the stratified prostaglandin or bile-synthesizing groups relative to their respective ‘Low PG’ or ‘Low Bile’ groups using *t*-tests to identify the statistical differences between normalized transcript levels in the phenotypes vs. the ‘Low PG’ or ‘Low Bile’ groups (Figure 3B, Figure 4, Figure 6B and Figure 7).

Kaplan–Meier survival curves were analyzed using the log-rank test, with statistical significance calculated from the chi-squared value compared to the ‘Low PG’ subgroups (Figure 4A) and the ‘Low Bile’ subgroups (Figure 6A).

In Appendix A, the overlap between the stratified prostaglandin and bile tumors is compared with previously identified tumor subtypes; the tumors were stratified with respect to Treg infiltration and HIF markers or were based on tumor sex-steroid generation [1]. Chi-squared tests were also used to analyze the statistically significant overlap between the groups.

Excel was used to perform the chi-squared tests, the *t*-tests, and the Kaplan–Meier log-rank tests.

For the cross-correlation analysis, shown in Appendix A, the Gliovis platform was used to calculate Pearson’s correlation between the transcript pairs as indicated by the numeral and the superscripted stars (*); statistical significance is indicated when *p* < 0.05 *, *p* < 0.01 **, and *p* < 0.001 ***.

## Figures and Tables

**Figure 1 ijms-24-03198-f001:**
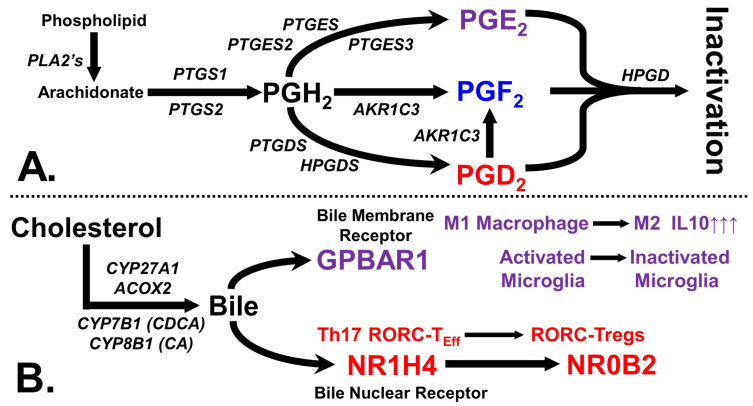
Enzymes and receptors used for prostaglandin/bile salt synthesis and signaling. (**A**) The synthesis and catabolism of prostaglandins is illustrated. (**B**) An illustration of the main enzymes utilized by the ‘acidic’ bile synthesis pathway and the resulting immunosuppressive effects via the plasma membrane-spanning G protein-coupled bile acid receptor 1 (GPBAR1) present on macrophages and microglia and the nuclear receptor subfamily 1, group H, member 4 (NR1H4) expressed by T cells.

**Figure 2 ijms-24-03198-f002:**
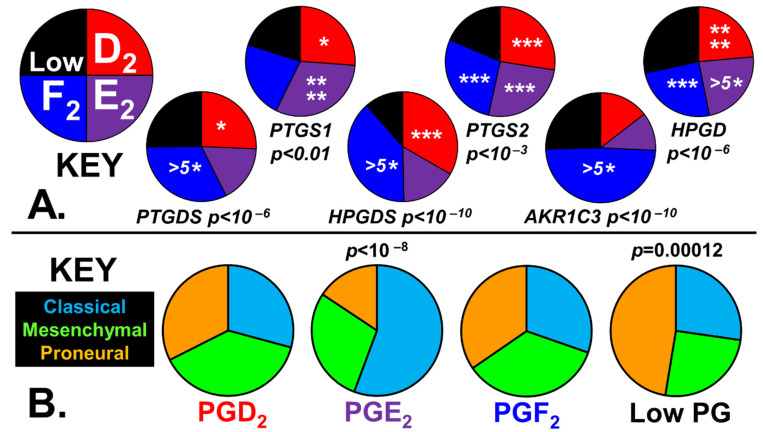
Phenotypes of GBM tumors are stratified into four prostaglandin-synthesizing subgroups. (**A**) The median expression levels of the enzyme transcripts used in the prostaglandin stratification are illustrated in the pie charts, and the significance derived from two different statistical tests is also indicated. The *p*-value calculated from the chi-squared statistic, comparing the transcript medians of the four groups with the total population, is below each chart. The stars within each slice indicate the statistical significance of transcript distributions in individual prostaglandin-synthesizing subgroups with respect to ‘Low PG’. Statistical significance from *t*-tests is represented by asterisks (*), *p* < 0.05 *, *p* < 0.005 ***, *p* < 0.001 ****, and *p* < 0.0001 ^(5^*^)^. (**B**) The fraction of the three canonical GBM subtypes within each of the PG phenotypes is represented in pie charts, with the chi-squared statistics indicated.

**Figure 3 ijms-24-03198-f003:**
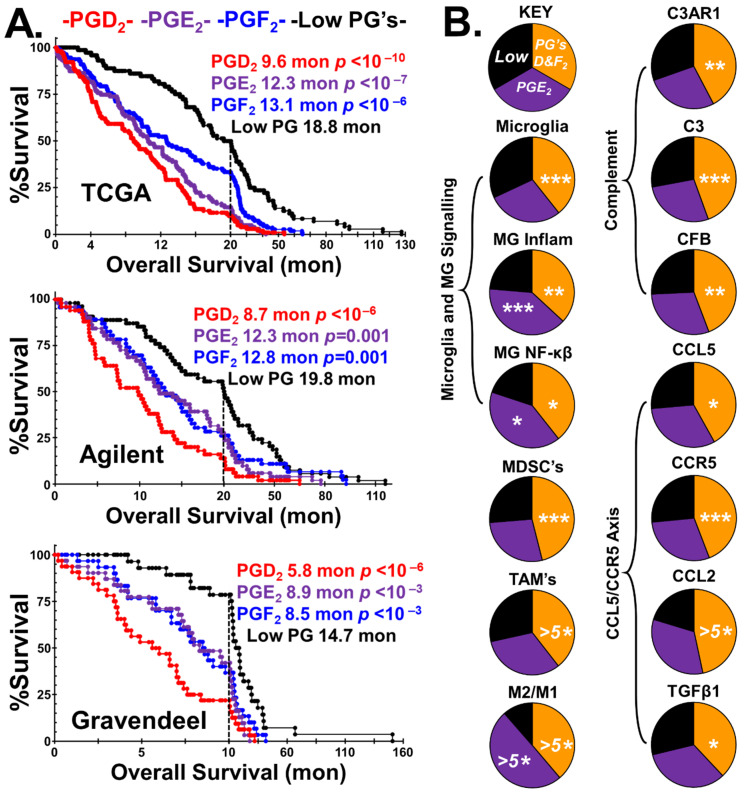
General characteristics of the prostaglandin groups. (**A**) The prostaglandin GBM tumor phenotype’s impact on patient outcome is shown in the Kaplan–Meier survival curves, with the median survival and statistical significance vs. the ‘Low PG’ group indicated. (**B**) The differences between the combined PGD_2_/PGF_2_ and PGE_2_ groups with the tumor/immune cell phenotype are highlighted. The PGD_2_/PGF_2_ group is enriched with microglia, MDSCs, and TAMs, with M2 > M1 polarization. The PGD_2_/PGF_2_ group is also enriched with markers of the CCL5/CCR5 axis. Statistical significance derived from the *t*-tests when compared to the ‘Low PG’ group are represented by asterisks, *p* < 0.05 *, *p* < 0.01 **, *p* < 0.005 ***, and *p* < 0.0001 ^(5^*^)^.

**Figure 4 ijms-24-03198-f004:**
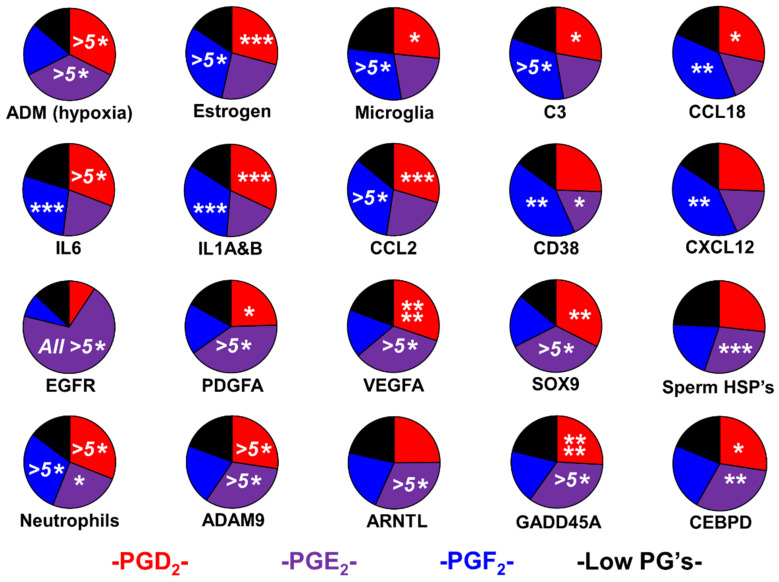
The main drivers and outputs for the four prostaglandin phenotypes are ‘Low PG’, PGD_2_, PGF_2_, and PGE_2_. The median transcripts of genes or gene baskets are displayed as color-coded pie charts with *t*-test significance levels represented by asterisks (*), *p* < 0.05 *, *p* < 0.01 **, *p* < 0.005 ***, *p* < 0.001 ****, and *p* < 0.0001 ^(5^*^)^. Hypoxia-regulated gene ADM shows that the PGD_2_ and PGE_2_ groups are hypoxic, while the PGF_2_ and ‘Low PG’ groups are normoxic. The PGD_2_- and PGE_2_-expressing tumors display elevated levels of estrogen signaling markers. Expression of interleukins IL6, IL1A, and IL1B are highly elevated in the PGD_2_ and PGF_2_ groups. The PGF_2_ group has enriched CD38 and CXCL12. The PGE_2_ groups show a bias toward the expressions of markers associated with the classical GBM phenotype, stem cell marker SRY-box transcription factor 9 (SOX9), and sperm-associated HSPs. Neutrophil infiltration of these tumors is high in all the prostaglandin-positive groups, but the activated neutrophil marker ADAM9 is only highly elevated in the PGD_2_ and PGE_2_ groups. PGE_2_-response genes aryl hydrocarbon receptor nuclear translocator-like protein 1 (ARNTL), growth arrest and DNA-damage-inducible protein GADD45 alpha (GADD45A), and CCAAT/enhancer-binding protein delta (CEBPD) correlate with tumor PGE_2_ synthesis. Neutrophil infiltration of the tumors is high in all the prostaglandin-positive groups. However, the expression of ADAM9 (a marker of activated neutrophils [84]) is only significantly elevated in the PGD_2_ and PGE_2_ groups.

**Figure 5 ijms-24-03198-f005:**
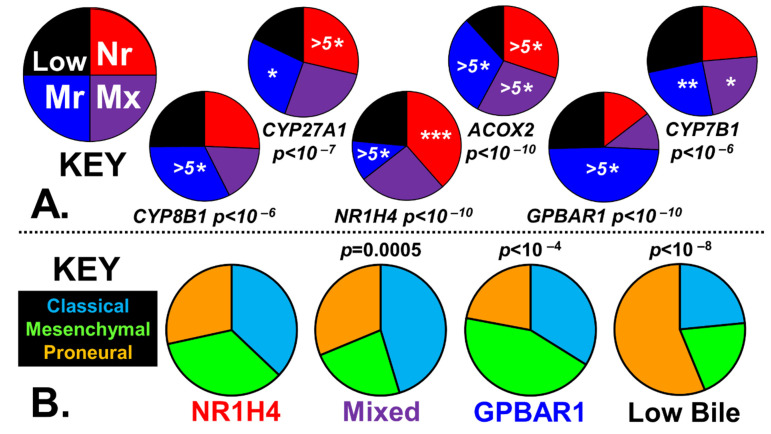
Phenotypes of GBM tumors are stratified into bile salt/receptor stratified phenotypes. (**A**) The median expression levels of the enzyme transcripts used for bile-generating/bile-receptor stratification are illustrated in the pie charts, with two statistical tests performed. Below each pie chart is the *p*-value calculated from the chi-squared statistic, which examines the distribution of transcript medians of the four groups compared to the total population. The stars within the pie chart slices indicate the statistical significance of the comparisons of transcript levels of the three individual bile-generating/bile-receptor synthesizing subgroups with respect to ‘Low Bile’. Statistical significance from the *t*-tests are represented by asterisks (*), *p* < 0.05 *, *p* < 0.01 **, *p* < 0.005 ***, and *p* < 0.0001 ^(5^*^)^. (**B**) Pie charts represent canonical GBM-subtype fractions within each bile phenotype, with the chi-squared statistics indicated.

**Figure 6 ijms-24-03198-f006:**
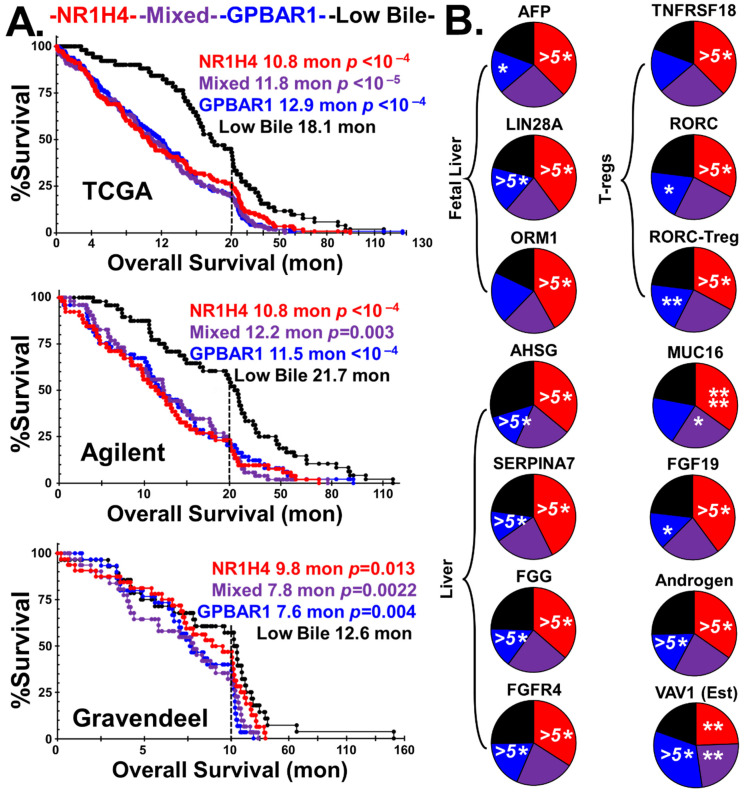
Bile salt impacts patient survival and the fetal liver phenotype of GBM. (**A**) Bile synthesis/receptor phenotypes’ impact on patient outcome is shown in the Kaplan–Meier survival analysis. The patient population with the worst outcomes are those that synthesize bile at > 80%, and those with low bile synthesis have better survival characteristics. (**B**) Pie charts show that the distribution of markers of the fetal and adult livers are predominantly in the NR1H4-rich tumors (Red). The NR1H4-rich tumors are infiltrated with RAR Related Orphan Receptor C (RORC)-rich Tregs and have elevated RORC-Treg target mucin 16 (MUC16), and high expression of the bile-sensitive hormone fibroblast growth factor 19 (FGF19). In addition, the NR1H4-rich tumors have an androgenic signature in contrast to the GPBAR1 receptor-rich tumors (Blue), which are estrogenic, clustering with estrogen reporter genes, such as vav guanine nucleotide exchange factor 1 (VAV1). *p* < 0.05 *, *p* < 0.01 **, *p* < 0.001 ****, and *p* < 0.0001 ^(5^*^)^.

**Figure 7 ijms-24-03198-f007:**
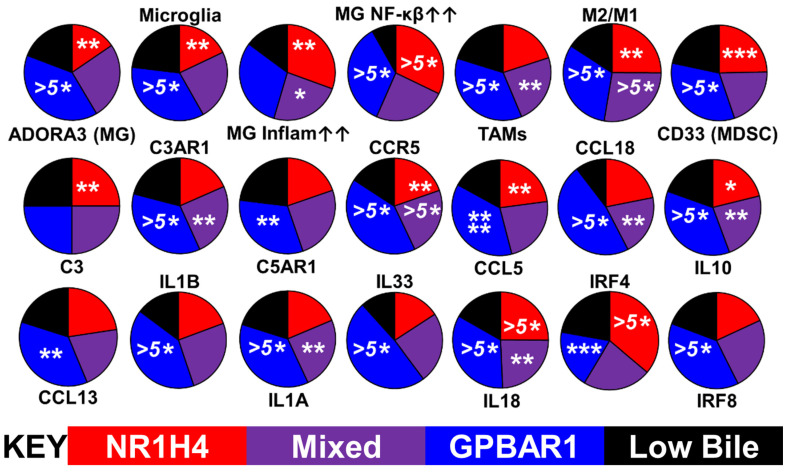
Association of GPBAR1 with the GBM tumor immune type. The GPBAR1-rich tumors are more heavily infiltrated with microglia, TAMs, and MDSCs, which display M2 >> M1 polarization markers. The GPBAR1-expressing subtype shows a distinct overlap with the microglial C3/C3AR1-driven PGD_2_/PGF_2_ phenotypes. CCL18, IL10, and CCL13. IL1B, and other members of the IL1 family, including IL33 and IL18, are elevated in the GPBAR1-rich tumors. Interferon regulatory factors 4 and 8 (IRF4 and 8) have a reciprocal relationship with the two bile acid receptors. The GPBAR1 tumors have high IRF8 expression, whereas levels are low in the NR1H4 tumors, but the expression of IRF4 is the opposite in these two tumor types. . Statistical significance from the *t*-tests are represented by asterisks (*), *p* < 0.05 *, *p* < 0.01 **, *p* < 0.005 ***, *p* < 0.001 ****, and *p* < 0.0001 ^(5^*^)^.

**Figure 8 ijms-24-03198-f008:**
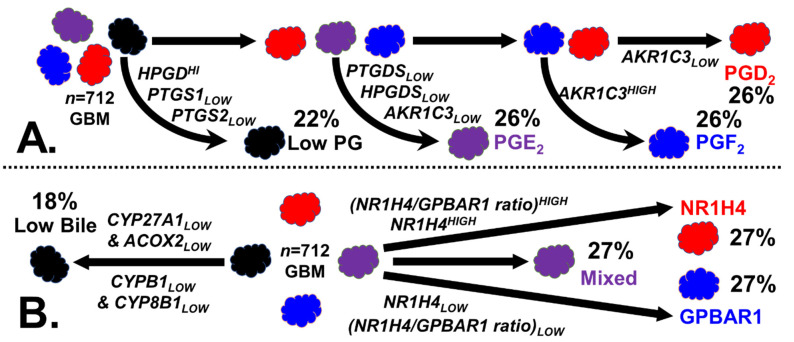
The procedure used for prostaglandin and bile GBM stratification. (**A**) How GBM tumors were stratified by their ability to generate prostaglandins is illustrated. Tumors incapable of generating prostaglandins are selected based on the PTGS1/2 and HPGD levels. Tumors incapable of synthesizing PGD_2_/F_2_, with low PTGDS, HPGDS, and AKR1C3, are placed in the PGE_2_ group. AKR1C3 levels are used to differentiate between the PGD_2_ and PGF_2_ groups. (**B**) Tumors under-expressing genes of the ‘acidic’ bile synthesis pathway are defined as ‘Low Bile’. Ratios of the two receptor transcripts are generated, and transcripts are ranked with respect to GPBAR1 and NR1H4. This procedure gives rise to three equally sized phenotypes: NR1H4 (GPBAR)1_LOW_, GPBAR1 (NR1H4)_LOW_, and, finally, ‘Mixed’, which is a group of tumors that express both receptors.

## Data Availability

All data sources and citations are to be found in the Section 4.

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
