# Peer review of "Acquisition of Immune Privilege in GBM Tumors: Role of Prostaglandins and Bile Salts"

_ijms, 2023, doi:10.3390/ijms24043198_

Round 1

Reviewer 1 Report

No comments

Author Response

Thank you for your generous review.

Reviewer 2 Report

The entire paper is rather convoluted and difficult to follow, especially for the general reader.

Specific comments:

1. Suggest to shorten the abstract to around 200 words as per the journal's guidelines to authors.

2. Please change "3 GBM transcriptome databases" to "three GBM transcriptome databases".

3. The central hypothesis that GBM tumors generate prostaglandins and bile salts to gain immune privilege is difficult to appreciate in relation to the broader literature, and it may not be a significant contributing mechanism. It is well-known that in humans, tumor-related chronic inflammation promotes an immunosuppressive microenvironment with a large number of immunosuppressive cells (M2 macrophages, MDSCs, Treg cells, etc.) and cytokines. This likely provides the main drive behind tumor occurrence, development, invasion and metastasis. I am not sure how the mechanisms uncovered in the present GBM transcriptome database analyses fit into the prevailing theories. e.g. what is the correlation between hypoxia and PGE2 and bile salts, TME and PGE2 and bile salts, Warburg effect and PGE2 and bile salts, etc?

4. Please make the writing more concise and succinct, sentences such as “Multiple pro-inflammatory neurological pathologies are ameliorated by GPBAR1 agonists, including multiple sclerosis[39], ischemia/reperfusion injury[40], subarachnoid hemorrhage[41], and traumatic brain injury[42]” do not seem to have much relevant to the discussion at hand and can be omitted.

5. Section 5.1 seems to be mostly lifted from a previous paper (see: pubmed.ncbi.nlm.nih.gov/34681642). Suggest to paraphrase or remove if not pertinent.

6. There is no need to capitalize ‘Self’.

7. Please change “immune priv-ilege” to “immune privilege”.

Author Response

Thank you for the generous review. 

Overall

The entire paper is rather convoluted and difficult to follow, especially for the general reader.

 We have used the services of Dr. Shaefali Rodgers and Jacob M. Kolman, MA, ISMPP CMPP, both senior scientific writers at the Houston Methodist Academic Institute, for a two-stage revision

The introduction is expanded and points in results have been placed in their proper place.

Specific comments:

  1. Suggest to shorten the abstract to around 200 words as per the journal's guidelines to authors.

Done

  1. Please change "3 GBM transcriptome databases" to "three GBM transcriptome databases".

Done

  1. The central hypothesis that GBM tumors generate prostaglandins and bile salts to gain immune privilege is difficult to appreciate in relation to the broader literature, and it may not be a significant contributing mechanism. It is well-known that in humans, tumor-related chronic inflammation promotes an immunosuppressive microenvironment with a large number of immunosuppressive cells (M2 macrophages, MDSCs, Treg cells, etc.) and cytokines. This likely provides the main drive behind tumor occurrence, development, invasion and metastasis.

"In the discussion we have added a paragraph to highlight other mechanisms known to drive immunosuppression

'Cancer cells can increase their survival by subverting the immune system in many ways, including the well-characterized ability of IDH-mutants to generate the immunosuppressive oncometabolite (D)-2-hydroxyglutarate and the elevation of immunosuppressive reactive oxygen species [112, 113].'"

I am not sure how the mechanisms uncovered in the present GBM transcriptome database analyses fit into the prevailing theories. e.g. what is the correlation between hypoxia and PGE2 and bile salts, TME and PGE2 and bile salts, Warburg effect and PGE2 and bile salts, etc?

We have prepared tables and performed statistical analysis on the overlap between different phenotypes.

“In Supplemental Tables 1–5 the overlap between stratified prostaglandin and bile tumors is compared with previously identified tumor subtypes, stratified with respect to Treg infiltration and HIF markers, or based on tumor sex-steroid generation [1]. Chi-square tests were also used to analyze the statistically significant overlap between groups.”

  1. Please make the writing more concise and succinct, sentences such as “Multiple pro-inflammatory neurological pathologies are ameliorated by GPBAR1 agonists, including multiple sclerosis[39], ischemia/reperfusion injury[40], subarachnoid hemorrhage[41], and traumatic brain injury[42]” do not seem to have much relevant to the discussion at hand and can be omitted.

This has been done with professional assistance.

  1. Section 5.1 seems to be mostly lifted from a previous paper (see: pubmed.ncbi.nlm.nih.gov/34681642). Suggest to paraphrase or remove if not pertinent.

It has been removed and we now focus on the main drive of the data.

  1. There is no need to capitalize ‘Self’.

All now in lower case.

  1. Please change “immune priv-ilege” to “immune privilege”.

Done

Reviewer 3 Report

I read carefully the manuscript entitled “Acquisition of Immuno-Privilege in GBM Tumors: Role of Prostaglandins and Bile Salts” by Sharpe et al. which is very important and interesting work investigating the role of Prostaglandins and Bile Salts on glioblastoma (GBM). I have some comments to be your work clear for readers:

1.      In the abstract part kindly add shortly the used methods in your study.

2.      Remove the subsections of the Introduction part.

3.      Add space between references and texts throughout the whole manuscript.

4.      In the Materials and Methods add the subsection title to the first paragraph (line 112).

5.      Before the result part add the used statistical analysis section.

6.      Add at the end of the manuscript conclusion part containing briefly the outcomes of your study and recommendations.

7.      The manuscript needs language and editing polishing.

8.      Avoid as possible the use of “we or our” throughout the whole manuscript also 141 reference is too many for original manuscript kindly decrease them as possible

Author Response

Thank you for the generous review.

 I have some comments to be your work clear for readers:

  1. In the abstract part kindly add shortly the used methods in your study.

It has been rewritten in light of your suggestions.

"Postulating that glioblastoma (GBM) tumors generate anti-inflammatory prostaglandins and bile salts to gain immune privilege, we analyzed 712 tumors in silico from three GBM transcriptome databases for prostaglandin and bile synthesis/signaling enzyme-transcripts markers. Pan-database correlation analysis was used to identify cell-specific signal generation and down-stream effects"

  1. Remove the subsections of the Introduction part.

Done

  1. Add space between references and texts throughout the whole manuscript.

Done

  1. In the Materials and Methods add the subsection title to the first paragraph (line 112).

Done

  1. Before the result part add the used statistical analysis section.

Done. All statistical methods are now in a single section at the end of the Methods section.

  1. Add at the end of the manuscript conclusion part containing briefly the outcomes of your study and recommendations.

Without any IP-issues we have added a pair of paragraphs.

"5.3. Diagnosis, Tracking, and Treatment of GBM with Vaccines

A large fraction of GBMs express testicular-specific proteins [1] and/or fetal liver-specific transcripts. Serum fetal liver proteins are routinely measured in hospitals to diagnose pregnancy-linked pathologies, i.e., α-fetoprotein or placental alkaline phosphatase and pregnancy-specific beta-2, 6, and 8. These validated tests could also be used in GBM patients for tracking tumor response to treatment and progression. The human fetal liver synthesizes and secretes biologically active chorionic gonadotropin. Provisional ELISA assays show that this pregnancy marker protein is present in the serum of both male and female GBM patients.

It has previously been demonstrated that GBM patients have circulating antibodies toward testicular-specific proteins [1]. The expression of fetal-specific proteins by GBM tumors opens an extensive range of new vaccination targets. Their efficacy will be im-proved if it is demonstrated that GBM patient serum has pre-existing antibodies toward fetal proteins. The advantage of targeting a fetal protein with an anti-cancer vaccine is the lower likelihood of off-target autoimmune responses"

  1. The manuscript needs language and editing polishing.

 We have used the services of Dr. Shaefali Rodgers and Jacob M. Kolman, MA, ISMPP CMPP, both senior scientific writers at the Houston Methodist Academic Institute, for a two-stage revision.

  1. Avoid as possible the use of “we or our” throughout the whole manuscript also 141 reference is too many for original manuscript kindly decrease them as possible

We have removed ‘We’ in all but three instances.

References have been cut back to 126.

Reviewer 4 Report

The authors submitted an interesting original paper, which deals with prostaglandins and bile salts in GBM tumours pathogenesis. Since GBM Tumours are a serious type of primary brain tumour, the paper topic is important and timely.

The manuscript is well written. The text shows a logic structure and the experimental data as well as the results are illustratively presented. The used methods are adequate and the results support the conclusions.

Nevertheless, I do not like that the authors negotiated completely the role of reactive oxygen species (ROS) in the research although it is known that the ROS play an important role like secondary messages in immune system and transcription processes, included in the signal processes involved in the prostaglandin cascade. Therefore, it fact should be at least briefly mentions in the introduction.

Additionally, for the future research, I would recommend searching for oxidative stress markers such as thiobarbituric acid reactive species, hydroxylated aromatic amino acids or other markers.

Author Response

Thank you for the generous review.

Nevertheless, I do not like that the authors negotiated completely the role of reactive oxygen species (ROS) in the research although it is known that the ROS play an important role like secondary messages in immune system and transcription processes, included in the signal processes involved in the prostaglandin cascade. Therefore, it fact should be at least briefly mentions in the introduction.

You are right about the importance of ROS in immunosuppression. However, we don’t have a clear correlation between ROS-proxies, like NRF2 downstream products NQO1 and GCLM, and prostaglandins or bile receptors. If anything, NQO1 and GCLM correlate with estrogen synthesis genes/estrogen reporters. We didn’t want to add another layer of complexity. In the light of your very valid point we added this text:-

“Cancer cells can increase their survival by subverting the immune system in many ways, including the well-characterized ability of IDH-mutants to generate the immunosuppressive oncometabolite (D)-2-hydroxyglutarate and the elevation of immunosuppressive reactive oxygen species [112, 113].”

Additionally, for the future research, I would recommend searching for oxidative stress markers such as thiobarbituric acid reactive species, hydroxylated aromatic amino acids or other markers.

You are right here. I also look at aldehydes and ketones.

Round 2

Reviewer 2 Report

Thank you for the revisions.

Reviewer 3 Report

The authors established all the required corrections and I have not any more comments